# Evaluation of Different Bacterial Wilt Resistant Eggplant Rootstocks for Grafting Tomato

**DOI:** 10.3390/plants10010075

**Published:** 2021-01-01

**Authors:** Ravishankar Manickam, Jaw-Rong Chen, Paola Sotelo-Cardona, Lawrence Kenyon, Ramasamy Srinivasan

**Affiliations:** World Vegetable Center, P.O. Box 42, Tainan 74199, Taiwan; jaw-rong.chen@worldveg.org (J.-R.C.); paola.sotelo@worldveg.org (P.S.-C.); lawrence.kenyon@worldveg.org (L.K.); srini.ramasamy@worldveg.org (R.S.)

**Keywords:** *Ralstonia*, *Solanum lycopersicum*, *Solanum melongena*, grafting-compatibility, fruit

## Abstract

Bacterial wilt (BW) is one of the most economically important diseases of tomato and eggplant in the tropics and subtropics, and grafting onto resistant rootstocks can provide an alternative and effective solution to manage soil-borne bacterial in these crops. This study was conducted to evaluate the BW resistance and agronomic potential of newly identified eggplant accessions as rootstocks for tomato grafting. Five BW resistant eggplant accessions (VI041809A, VI041943, VI041945, VI041979A, and VI041984) from the World Vegetable Center were evaluated as rootstocks for grafting with two different fresh market tomato cultivars (Victoria and TStarE) as scion under open field conditions in Taiwan. Graft compatibility using the tube grafting method as well as BW wilting percentage, disease index, fruit yield and quality parameters were assessed. All the rootstocks showed good graft compatibility (93% and above) and grafted plants showed low wilting percentage (0.0–20.0%) and disease index (0.0–20.8%) following inoculation with BW. Yield for the eggplant rootstock grafted tomato plants was higher compared to the non-grafted tomatoes and self-grafted tomato. Fruit quality was not affected by grafting, although some differences in antioxidant activities were observed. The new eggplant rootstocks can be considered as alternatives to the rootstocks currently used for commercial production of tomatoes during the hot-wet season.

## 1. Introduction

Tomato (*Solanum lycopersicum* L.) is ubiquitous and the most important fruit vegetable crop produced throughout the tropics and subtropics, and is an important source of nutrients such as vitamins A and C, antioxidants and carotenoids [1,2]. Production of high value fruits and vegetables such as tomatoes during the off-season offers small-holder farmers an opportunity to change from subsistence to commercial farming and substantially increase their incomes [3,4,5]. Unfortunately, among many diseases that affect tomato farming, tomato yellow leaf curl disease caused by whitefly-vectored begomoviruses [6,7], bacterial wilt caused by *Ralstonia* spp. [8,9], and late blight caused by *Phytophthora infestans* [10,11] are of serious concern during the off-season due to the favorable weather conditions (hot and wet) for these biotic constraints.

Bacterial wilt (BW), caused by soil-borne bacteria of the *Ralstonia solanacearum* species complex (RSSC) formally known as *Pseudomonas solanacearum* E.F Smith is one of the most economically important diseases of tomato and eggplant (*Solanum melongena* L.) in the tropics and subtropics, especially if the production is targeted for the higher value off-season market [12]. BW was first described by Smith [13] in potato (*Solanum tuberosum* L.), tomato, and eggplant and can cause up to 100% economic losses [14,15]. Severity of symptoms induced by BW, the wide geographic distribution of BW, and the broad host plant range which includes more than 200 plant species belonging to 53 different families [16,17] are some of the most important factors contributing to major yield losses.

Strains of the RSSC were initially sub-divided into five “races” based on host ranges, and into five “biovars” based on carbohydrate utilization [18,19]. Later, sequence analysis of the internal transcribed spacer (ITS) region of the 16S-23S rRNA gene, further divided the RSSC into four phylogenetic groups (“phylotypes” I to IV) which corresponded to the geographical origins of strains from Asia, America, Africa, and Indonesia, respectively [20]. The phylotypes were further separated into “sequevars” based on the partial endoglucanase (*egl*) gene sequences [21]. In 2014, the RSSC underwent further taxonomic revision with phylotype I from Asia and phylotype III from Africa being reclassified as *R. pseudosolanacearum*, phylotype II remaining as *R. solanacearum* and phylotype IV from Indonesia and Australia shifting to *R. syzygii* [22]. The bacteria of the RSSC enter the host plant roots through natural openings and wounds and multiply in the vascular system filling and blocking the xylem elements, and generally leading to a sudden wilting of the whole plant while still green, and eventually plant death [23]. Several methods have been used to control BW, including soil disinfection, soil amendment, biological and chemical controls, and resistant cultivars or rootstocks for grafting [24,25,26]. Nevertheless, it is very difficult to manage BW as the pathogen survives many years in soil without host plants [27]. Chemical control is not effective due to the localization of the pathogen inside the plants specifically in the xylem vessels [9]. Antibiotics such as penicillin, ampicillin, tetracycline, and streptomycin have been reported to be less efficient in suppressing *R. solanacearum* growth particularly in open fields when compared with resistant cultivar, but also antibiotics are not recommended due to their potentially harmful effects to the environment and human health, and the buildup of antibiotic resistance in the local environment [28].

Breeding for resistance to BW is still the most appropriate, economically, and environmentally promising strategy for controlling this pathogen [23]. Grafting onto resistant rootstocks also provides an alternative and effective solution to manage soil-borne bacterial and fungal pathogens in Solanaceous and Cucurbitaceous crops [29]. Tomato and eggplant rootstocks are used for grafting tomatoes worldwide. However, eggplant rootstocks are preferred over tomato rootstocks in many parts of the world due to their stronger resistance to BW and tolerance to flooding [30]. In spite of these advantages, availability of eggplant rootstocks with BW resistance is limited. By screening the eggplant germplasm accessions of the Word Vegetable Center (WorldVeg) in earlier studies (unpublished data), a few eggplant accessions with BW resistance were identified. Thus, the present study was conducted to evaluate the BW resistance, as well as the agronomical potential, and the efficiency of the newly identified eggplant accessions in the field as rootstocks for tomato grafting.

## 2. Results

### 2.1. Graft Compatibility

The survival of grafted plants was between 93 and 100 percent in the first season, and 100 percent in the second season. There was a significant difference in graft compatibility between treatments (rootstocks) during the first season when the scion Victoria was used (F_10,31_= 4.31; Pr > F = 0.0023, Table 1). Most of the new rootstocks showed 95–100% compatibility except VI041945 (93%), and they did not differ from the current rootstock check VI045276. During the second season, there was no significant difference among all treatments including self-grafted tomatoes.

### 2.2. Wilt Percentage, Disease Index and Field Plant Survival

The eggplant rootstocks showed significant differences in wilting percentage in both seasons (2018: F_11,35_ = 38.39; Pr > F = < 0.0001; 2019: F_11,35_ = 39.84; Pr > F = < 0.0001). More specifically, the highest wilting was observed in the accession VI046095 (susceptible check) during both seasons, followed by tomato non-grafted and tomato self-grafted. In contrast, the lowest wilting was observed in the remaining accessions, which was consistent across the two seasons (Table 1). No significant difference was observed between the new rootstocks and the resistant rootstock check (VI045276) in wilting percentage with the range of 0.0 to 20.0 percent in both seasons.

Disease index also was significantly different for the different accessions in both seasons (2018: F_11,35_ = 27.92; Pr > F = < 0.0001; 2019: F_11,35_ = 31.23; Pr > F = < 0.0001). The highest disease index was observed in the accession VI046095 (susceptible check) during both seasons, followed by tomato non-grafted and tomato self-grafted compared to the remaining accessions (Table 1). More specifically, the DI for the susceptible check was significantly different from the DI of either of the two tomato accessions in 2018, but no significant differences in DI were observed among these accessions in 2019. There was significant difference in DI between new rootstocks (0.0 to 4.2) and the resistant rootstock check (4.2 to 8.3) evaluated in both the seasons. In general, all the newly identified eggplant rootstocks had reduced wilting percentage and DI compared to the other accessions in both seasons.

Field plant survival was significantly different among accessions in the two growing seasons (2018: F_11,35_ = 40.49; Pr > F = < 0.0001; 2019: F_11,35_ = 6.04; Pr > F = 0.0001) (Table 1). The field survival ranged from 11.5 to 12 during first season among the 12 plants transplanted per replication. However, during the second season the plant survival ranged from 8.5 to 10.8, though there was no difference observed compared to resistant check (8.5). In both seasons all of the susceptible check died. There was no significant difference in field survival between self-grafted and non-grafted tomatoes in either seasons (Table 1).

### 2.3. SPAD Value, Fruit Yield, Marketable Fruit Weight, Fruit Length, Fruit Width and Fruit Length to Width Ratio

There were no significant differences observed among all the treatments in the chlorophyll content (SPAD value) measured during the first growing season (F_10,31_ = 2.16; Pr > F = 0.0659). Chlorophyll content was not measured for the second season as the leaves were severely chlorotic due to yellow leaf curl disease. Total fruit yield was highly significantly different in the first season (2018: F_11,35_ = 15.05; Pr > F = < 0.0001), whereas during the second season the plants were affected by yellow leaf curl disease that resulted in insignificant yield among the treatments. The highest total yield was recorded in the accessions VI041809A, VI041943, and VI041979A (Table 2). The highest per plant yield was recorded in the non-grafted tomato (Figure 1). The resistant rootstock check did not show significant difference compared to the other treatments. There were no fruits harvested from the susceptible check as all the plants wilted completely.

Marketable individual fruit weight showed significant differences (2018: F_11,35_ = 62.30; Pr > F = < 0.0001) among the accessions (Table 2), which followed a similar trend as the total fruit yield; the highest fruit weight was recorded from the accessions VI041809A, VI041943, and VI041979A along with self-grafted and non-grafted tomatoes. The fruit weight data for the second season was not recorded as the fruit yield was very low. Fruit length was significantly different among the accessions (F_10,31_ = 2.81; Pr > F = 0.0222). The lowest fruit length was recorded for the accession VI041945, whereas all other treatments recorded the highest fruit length (Table 3). No significant differences were observed on fruit width (F_10,31_ = 2.57; Pr > F = 0.0331) and fruit length to width ratio (F_10,31_ = 1.42; Pr > F = 0.2300) among all the accessions.

### 2.4. Fruit Quality Parameters (pH, Soluble Solid, Acidity, Color, β-Carotene, Lycopene, Antioxidant Activity)

The pH of the harvested fruits was significantly different for the accessions (F_10,31_ = 3.24; Pr > F = < 0.0112). The lowest pH value was recorded in the accession VI041945. There were no significant differences among the new rootstocks and the resistant rootstock check.

Total soluble solid in the fruits were significantly different for the accessions (F_10,31_ = 12.25; Pr > F = < 0.0001). The highest content was observed in accessions VI041945 and VI041984 (Table 3). All other treatments including grafted and non-grafted tomatoes recorded significantly lower total soluble solids. Significant differences were observed in the antioxidant content among the accessions (F_10,31_ = 6.29; Pr > F = < 0.0001). The highest antioxidant content was recorded in VI041945. Whereas, there were no significant differences for the other fruit quality parameters: acidity (F_10,31_ = 2.23; Pr > F = 0.0585), Color (F_10,31_ = 0.98; Pr > F = 0.4891), β-carotene (F_10,31_ = 2.08; Pr > F = 0.0755), and lycopene (F_10,31_ = 1.25; Pr > F = 0.3180).

## 3. Discussion

Graft compatibility is one of the most important criteria for developing new rootstocks for vegetable grafting [31]. The tomato cultivars used in this study were highly compatible with all the tested eggplant rootstocks. Many other authors have reported that eggplant cultivars or accessions can be used as rootstocks for tomato grafting [32,33]. The grafting compatibilities of the five newly identified eggplant accessions were similar to that of the currently widely used rootstock and resistant check (EG203) in this study. The BW wilting percentage and disease index of the newly identified rootstocks were similar to those of EG203, and significantly lower than for the susceptible control (VI046095), indicating that the new rootstock accessions have good field resistance to BW and could be used in place of EG203. There was no difference in wilting or DI between self-grafted and non-grafted tomato plants, indicating that the grafting process is not the cause of observed resistance. The field survival of the plants is the direct effect of the disease resistance by reduced wilt and disease index.

Grafting on the new/resistant eggplant rootstocks gave greater yield/plot compared to grafted plants on susceptible eggplant or tomato or non-grafted plants because the plants grafted on resistant eggplant were less or unaffected by BW in the field. This shows that the advantage of grafting tomato with resistant rootstock is in increased plant survival combined with yield advantages compared to non-grafted tomatoes [34,35]. However, if there is no disease pressure in the field, there will be no significant advantage in grafting on to eggplant rootstock. This was in line with a meta-analysis indicating that among the 949 combinations of grafted tomatoes reviewed, only 37% exhibited higher yields than non-grafted tomatoes [34]. However, some grafting experiments with robust rootstocks such as F1 hybrid rootstocks resulted in higher yield compared to non-grafted plants by mobilizing more nutrients to the scion [36,37]. There is no reduction in individual fruit weight or size in our study, which is in line with Qaryouti [38], but some studies had reported reduction in fruit size of fresh market tomatoes due to interspecific grafting of tomatoes with eggplant [39] or *Solanum torvum* [40], which is the undesirable character for fresh market tomatoes apart from the reduction in the yield.

There are many reports of fruit quality improvement or reduction due to interspecific grafting. On the other hand, in the current study, two rootstocks (VI041945 and VI041984) showed an increase in total soluble solid content of the fruits, compared to other rootstocks and non-grafted tomatoes. This was also reported by many authors [41,42,43]. In contrast to this, Turhan [44] reported the reduction of soluble solids due to grafting. Thus, it is evident that the total soluble solids are influenced by the rootstock used for grafting. This was also supported by other research on tomato [38]. There is no influence of rootstocks on the color of the fruit, β-carotene or lycopene contents. It is evident from our experiment that most of these nutrients are not influenced by the rootstocks used in our study. Some earlier studies have shown contrasting evidence, with positive, or negative effect due to grafting [38], or lack of influence [45]. The antioxidant activity was reduced in tomatoes with treatments involving VI041809A, VI041943, and VI041979A as rootstocks. This was also reported by Vrcek [46], but there is no influence of rootstock on the antioxidant activity in the other two rootstocks tested (VI041945 and VI041984). Hence, the antioxidant activity also varies with the specific eggplant rootstocks used for grafting.

## 4. Materials and Methods

### 4.1. Plant Materials and Experimental Location

All the experimental lines (Table 4) were obtained from Genetic Resources and Seed Unit (GRSU), Genebank of WorldVeg in Shanhua, Taiwan (23°06′53.1″ N, 120°17′53.5″ E). These accessions were earlier identified as bacterial wilt resistant by initial screening under net house conditions, and subsequently the accessions were purified and bulked up through single-seed descent and the BW resistance of the progeny confirmed. Tomato variety “Victoria” (*Known-you seed* Co., Ltd., Kaohsiung, Taiwan) was used as scion in the first season and “TStarE” from the Tomato Breeding group at WorldVeg was used as scion in the second season trials. “TStarE” supposed to be resistant to Tomato Leaf Curl Virus, was used in second trial to prevent the viral diseases during that season. The accession VI045276, popularly known as EG203 and widely used rootstock in many developing countries was used as resistant check [26,47], and the accession VI046095, popularly known as EG048 was used as BW susceptible check [48].

The field experiments were conducted during 2018 and 2019 for two different seasons at the experimental farm of the WorldVeg, Shanhua, Taiwan (23°06′30.5″ N, 120°17′49.9″ E, 8 m a.s.l.). Accumulated annual rainfall usually approaches 2000 mm, of which more than eighty percent falls during the rainy months of May to September. The mean total precipitation that occurred during the experimental growth periods in first season (October 2018–January 2019) and second season (April–June 2019) was 5.00 mm and 2404.00 mm, respectively. Intermittent irrigation was applied whenever required. Weather parameters of the study period are given in Table 5. The soil type was sandy loam (18% clay containing illite (non-expanding clay mineral) and vermiculite, 27% silt, 55% sand) with low total-N content (<0.5%) and a pH of about 7.

### 4.2. Seedling Production, Grafting and Pathogen Inoculation

The seedlings were raised using a commercial potting media consisting of cocopeat and peat moss in 72-cell trays with each cell measuring 40 × 20 × 45 mm with the cell capacity of 40 mL. Both the rootstocks and scion plants were used for grafting after reaching the three true leaf stage. The seedlings were watered by hand and a 17-17-17 NPK fertilizer solution diluted to 2.5 g/L was applied each week. The scions were three-week-old tomato seedlings with two to three true leaves, which were grafted onto five-week-old eggplant rootstocks with two to three true leaves [49]. The tube-splice-grafting method [50] was used for grafting. Briefly, the stems of the scions and rootstocks were cut obliquely at a 30° angle above the cotyledons using sterile blades. The surfaces of the cut scions and rootstocks were then gently joined and held together within a latex tube of 2.0 mm-inner diameter and length of about 12 mm. The grafted plants were kept in a dark healing chamber for three days, then kept to recover for 10 days in a greenhouse where they were progressively exposed to more sunlight each day. Three days before transplantation the roots of the seedlings were wounded with a knife and inoculated with 10 mL plant of a mixed suspension of *Ralstonia pseudosolanacearum* strains Pss4 and Pss97 containing 108 cfu/mL. The mixed suspension was used to evaluate against the two widespread strains. Pss4 is a phylotype I (*R. pseudosolanacearum*), race 1, biovar 3 strain, endemic to Taiwan and isolated from tomato. Pss4 is highly aggressive [51] and has been used for routine screening and breeding for bacterial wilt resistance at WorldVeg for many years [52]. Pss97 is also Phylotype I, race 1, biovar 3, but isolated from an infected eggplant from Pingtung County of Southern Taiwan in 1991. Both Pss4 and Pss97 belong to the predominant virulence group in Taiwan. The bacterial strains were grown on plates 523-medium [53] at 30 °C for 24 h, harvested with glass slides, suspended in water, and adjusted to OD600 = 0.3 (about 108 cfu per mL). Grafted plants were transplanted to the field making sure that the graft union was above the soil surface to avoid the development of adventive roots from the scion.

### 4.3. Experimental Design

A randomized complete block design (RCBD) was used for the grafting compatibility and field experiments. For the grafting compatibility experiments, 18 plants per treatments were used and replicated for four times. Self-grafting of tomato scion onto tomato were included as check. After assessing their graft compatibility, the seedlings were inoculated with *Ralstonia pseudosolanacearum* three days before transplantation. The field experiments were conducted in the bacterial wilt sick plot (WorldVeg field no: 53) where each plot was 2.7 m × 1.5 m and the replications were separated by 2.0 m. The beds were covered with 15-micron silver-black plastic mulching sheet before transplanting the seedlings. Twelve seedlings per plot were planted through holes in the mulching sheet in two rows of six plants (50 cm between rows, 45 cm between plants within a row) and the experiment was replicated for four times. Suckers (side shoots from rootstock) that developed on the eggplant rootstocks near the cotyledons were removed. The plants were trained up bamboo stakes, and standard practices for irrigation, fertilizer application, pest control, and pruning were followed.

### 4.4. Graft Compatibility

One week after grafting, grafting compatibility was assessed. Successfully grafted plants were those that had produced new leaves, whereas the graft-failed plants were those that had wilted and had not produced new leaves. The number of successfully grafted plants for each replication was counted and the percentage of successful grafted plants were calculated.

### 4.5. Wilting Percentage (W%) and Disease Index (DI)

At four weeks after inoculation (25 days after transplanting to the field) each plot was assessed for percentage of plants wilting and the disease severity for each plant was scored using the zero-to-five rating scale [54] where 0 = no symptoms, 1 = 1 leaf partially wilted, 2 = 2 or 3 leaves wilted, 3 = all except the top 2 or 3 leaves wilted, 4 = all leaves wilted and 5 = dead. From the disease severity scores the disease index (DI; %) for each plot was calculated using the formula: DI = [(N0 × 0 + N1 × 1 + N2 × 2 + N3 × 3 + N4 × 4 + N5 × 5)/(Nt / 5)] × 100, where N0 to N5 = number of plants with disease rating scale values from 0 to 5, and Nt = total number of plants. The resistance reaction of accession was based on the W% and DI at the fourth week after inoculation (WAI), and categorized by DI at the fourth WAI. Accessions with DI from 0% to 20% were considered resistant (R), between 20% and 40% were moderately resistant (MR), between 40% and 60% were moderately susceptible (MS), and over 60% were regarded as susceptible (S) [55].

### 4.6. Chlorophyll Content

A portable chlorophyll meter (SPAD-502, Minolta Camera Co. Ltd., Osaka, Japan) was used for rapidly and nondestructively assessing foliar N status of the grafted plants in the field during peak harvest stage for relative comparison purposes. The fully opened third leaf from the top was chosen for the reading with a measurement area of 2 mm × 3 mm. Three measurements were taken for each plant and the averages were computed. All the survived plants in the entire treatment were observed for the SPAD values for each replication. Second season leaves were affected by begomovirus yellow leaf curl diseases and hence the readings for second season were not considered.

### 4.7. Fruit Yield

The fruits were harvested at the appropriate stage of ripeness, and the marketable and unmarketable fruits were separated plot wise and weighed. Fruit numbers were counted. Fruit length and width at mid-length were measured during the peak harvest. In both experimental years, fruits at the full-red stage were harvested from each plot once when one or more clusters on most plants within plots had ripe fruits.

### 4.8. Fruit Quality

Fruit quality was measured for the fruits collected during the peak harvest i.e., 4th harvest out of 7 during the first season. The second season trial was affected by begomovirus yellow leaf curl disease, so the fruit quality was not measured.

#### Sample Preparation

Each sample consisted of >600 g fully ripened fruits harvested from a single plot. Fruits were cut, blended in a homogenizer and filtered through gauze to remove seeds, skin, and membranes. From each sample, six plastic bags were prepared, each containing 10–20 g of tomato slurry. The bags were sealed and immediately stored at −70 °C for subsequent analyses of carotenoids, and citric acid. Supernatants obtained after centrifugation at 6000× *g* for 10 min were used on the same day to measure color, and soluble solids concentration (SSC). Standard methods were used for analysis of total soluble solid, β-carotene, lycopene, and antioxidant activity [56]. Titratable acidity (TA) was measured by mixing 2 g of pulp and 50 mL distilled water with a few drops of phenolphthalein indicator and titrating the mixture with 0.1 N NaOH. The titratable acidity was expressed as percentage citric acid equivalent to the quantity of NaOH used for the titration [57]. The ratio of SSC to titratable acidity was calculated.

### 4.9. Statistical Analysis

Data was checked for normality using the Shapiro–Wilk test [58] (Proc UNIVARIATE, SAS). Percentage and incidence data were arcsine transformed (*arcsin* (*sqrt*(*x*)). Data was then analyzed using Proc GLM of SAS, version 9.4 (SAS Institute, Cary, NC, USA). Significant differences were identified; means were separated by Tukey’s Honestly Significant Difference (HSD) post hoc test (differences were considered significant at α = 0.05). Non-transformed means were used in the results section.

## 5. Conclusions

Tomato growers in Taiwan and Vietnam have been using the bacterial wilt resistant eggplant rootstock (EG203) developed by World Vegetable Center for grafting tomatoes especially during the hot-wet season for many years. One of our studies found that because of 100% adoption of tomato grafting in Lam Dong province in Vietnam, the estimated total profit for tomato farmers was US $41.7 million higher than if the same area had been planted with non-grafted tomatoes. Recently in some areas this rootstock appears not to be as effective as it used to be. This study identified five new eggplant rootstocks that can be used as alternatives to the rootstocks currently being used in countries such as Taiwan and Vietnam for commercial production of tomatoes during the hot-wet season. There is no difference in yield when compared to the resistant rootstock check, though there is a reduction in fruit yield per plant. Nevertheless, the advantage is the field plant survival during high disease pressure. There was little difference in the nutrient content of the fresh market tomatoes when grafted with eggplant rootstocks and it was mainly influenced by the specific rootstock scion combinations. The results of this study provide the opportunity for the farmers to broaden the diversity of rootstocks they use and determine if the BW resistance of EG203 is compromised in some areas and EG203 may better be replaced by one of these newly identified bacterial wilt resistant rootstocks. Hence, there is an opportunity to the farmers in Asia and sub-Saharan Africa to widen the availability of the rootstocks and option to replace old rootstocks if the bacterial wilt resistance breaks down. Further studies are needed to confirm the field performance and resistance to BW across different countries and also with different fruit types such as cherry tomatoes.

## Figures and Tables

**Figure 1 plants-10-00075-f001:**
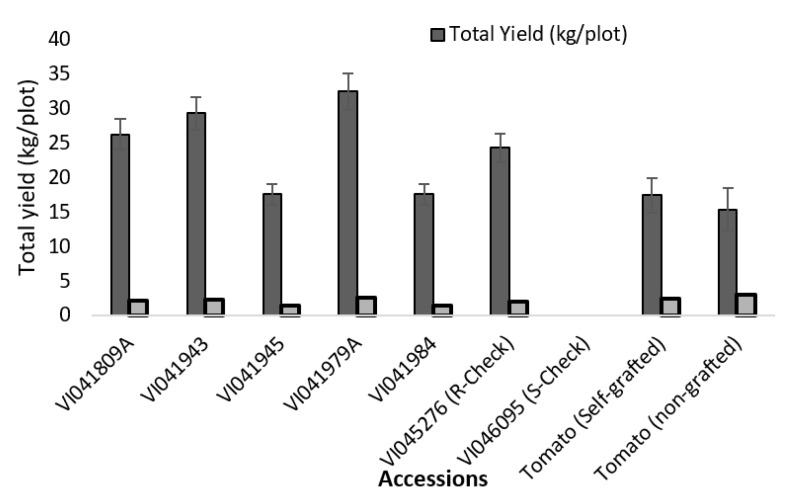
Comparison of total yield per plot versus yield per plant during the first season.

**Table 1 plants-10-00075-t001:** Graft compatibility of different rootstocks and effect of rootstock accession on wilting %, BW disease index (DI) and field survival of grafted plants.

Accession	Graft Compatibility (%) *		Wilting (%)		Disease Index (%)		Field Survival (out of 12 Plants)	
	2018	2019	2018	2019	2018	2019	2018	2019
VI041809A	100 a	100	2.1 c	10.4 c	2.1 b	10.4 b	11.8 a	9.5 a
VI041943	100 a	100	0.0 c	5.8 c	0.0 b	8.3 b	12.0 a	10.8 a
VI041945	93 b	100	0.0 c	14.6 c	0.0 b	16.7 b	12.0 a	5.5 abc
VI041979A	96 ab	100	2.1 c	7.9 c	0.8 b	10.4 b	12.0 a	9.0 ab
VI041984	99 ab	100	2.1 c	20.0 c	2.1 b	20.8 b	11.8 a	8.8 ab
VI045276 (R-Check)	94 ab	100	4.2 c	7.5 c	4.2 b	8.3 b	11.5 a	8.5 ab
VI046095 (S-Check)	99 ab	100	100.0 a	87.5 a	100.0 a	91.7 a	0.0 c	0.0 c
Tomato (Self-grafted)	100 a	100	50.0 b	100.0 a	43.8 a	100.0 a	7.0 b	2.0 c
Tomato (non-grafted)	-	-	72.9 b	97.5 a	63.3 a	100.0 a	5.0 b	2.3 bc

* Values followed by a different lower-case letter(s) within a column are significantly different at the 5% probability level.

**Table 2 plants-10-00075-t002:** Yield parameters of the grafted tomato plants.

Accession	SPAD Value 2018	Marketable Fruit/Plot ^a^ 2018		Mean Fruit Weight (g)	Mean Fruit Length (mm)	Mean Fruit Width (mm)	Fruit (L:W)	Total Yield (kg/plot)	Total Yield (kg/plot)
		Weight (kg)	Number of Fruits/Plots	2018	2018	2018	2018	2018	2019
VI041809A	45.78 ± 2.02	17.62 ± 2.65	280 ± 42	62.9 ± 0.65 a	54.4 ± 1.69 a	47.6 ± 1.29	0.88	26.29 ± 5.37 ab	0.71 ± 0.71
VI041943	46.93 ± 0.63	17.07 ± 2.38	285 ± 35	59.9 ± 1.86 ab	53.9 ± 1.14 a	47.5 ± 1.76	0.88	29.36 ± 5.24 a	1.27 ± 0.87
VI041945	47.85 ± 2.68	11.42 ± 2.45	226 ± 28	50.5 ± 5.19 bc	48.3 ± 2.64 b	44.5 ± 2.51	0.92	17.61 ± 2.9 bc	1.01 ± 0.48
VI041979A	45.03 ± 1.2	19.32 ± 3.20	318 ± 53	60.9 ± 0.71 a	53.3 ± 1.28 ab	47.5 ± 2.07	0.89	32.49 ± 41 a	0.65 ± 0.33
VI041984	45.57 ± 1.67	11.24 ± 0.97	235 ± 15	47.8 ± 1.08 c	51.3 ± 3.62 ab	45.1 ± 2.75	0.88	17.66 ± 2.43 bc	0.92 ± 0.58
VI045276 (R-Check)	48.04 ± 1.94	15.80 ± 2.84	284 ± 40	55.5 ± 5.97 abc	53.6 ± 1.65 ab	46.7 ± 1.25	0.87	24.33 ± 4.04 abc	0.83 ± 0.52
VI046095 (S-Check)	-	0.0 ± 0.00	-	-	-	-	-	0.0 ± 0.00 d	0.29 ± 0.58
Tomato (Self-grafted)	46.28 ± 1.53	8.45 ± 4.35	136 ± 75	64.3 ±6.45 a	55.1 ± 2.36 a	48.3 ± 0.79	0.88	17.49 ± 7.77 bc	0.00 ± 00
Tomato (non-grafted)	49.03 ± 2.55	8.25 ± 2.71	131 ± 47	63.6 ±6.77 a	53.9 ± 2.88 a	47.8 ± 2.08	0.89	15.44 ± 4.69 c	0.07 ± 0.13

^a^ Fruit (L: W) length to width ratio; Values followed by a different lower-case letter(s) within a column are significantly different at the 5% probability level.

**Table 3 plants-10-00075-t003:** Effect of different eggplant rootstocks on fruit quality parameters of grafted tomatoes. SS = soluble solid (SS); acidity; color; β-Carotene; lycopene; antioxidant activity (AA).

Accession	pH *	SS (°Brix)	Acidity (% Citric Acid)	Color (a/b)	β-Carotene (mg/100g)	Lycopene (mg/100g)	AA (μmole TE/100g)
VI041809A	4.14 a	4.95 b	0.32 b	1.25	0.49	5.71	326.15 b
VI041943	4.10 ab	5.00 b	0.34 ab	1.25	0.53	5.93	350.35 b
VI041945	4.04 b	6.15 a	0.36 ab	1.25	0.52	6.57	437.67 a
VI041979A	4.16 a	4.90 b	0.33 ab	1.28	0.43	5.82	320.25 b
VI041984	4.11 ab	6.10 a	0.37 ab	1.28	0.53	6.97	393.77 ab
VI045276	4.10 ab	5.30 b	0.35 ab	1.26	0.54	6.01	389.85 ab
VI046095	-	-	-	-	-	-	-
Self-grafted Tomato	4.17 a	4.80 b	0.35 ab	1.25	0.48	6.22	382.71 ab
Non-grafted Tomato	4.16 a	5.00 b	0.40 a	1.18	0.48	5.21	379.36 ab

* Values followed by a different lower-case letter(s) within a column are significantly different at the 5% probability level.

**Table 4 plants-10-00075-t004:** Accessions from *Solanum melongena* and *Solanum lycopersicum* used as rootstock or scion for the evaluation of bacterial wilt (BW).

WorldVeg Accession Number or Cultivar Name	Rootstock/Scion	Species	Character	Origin
VI041809A	Rootstock	*Solanum melongena*	BW resistant	India
VI041943	Rootstock	*S. melongena*	BW resistant	India
VI041945	Rootstock	*S. melongena*	BW resistant	India
VI041979A	Rootstock	*S. melongena*	BW resistant	India
VI041984	Rootstock	*S. melongena*	BW resistant	India
VI045276 (EG203)	Rootstock	*S. melongena*	BW Resistant check	India
VI046095 (EG048)	Rootstock	*S. melongena*	BW Susceptible check	Denmark
Victoria	Scion	*Solanum lycopersicum*	Fresh market Tomato	Taiwan
TStarE	Scion	*S. lycopersicum*	Fresh market Tomato	Taiwan

**Table 5 plants-10-00075-t005:** Weather parameters during the field experiments for two different seasons at the experimental farm of the WorldVeg, Shanhua, Taiwan.

Season	First Season	Second Season
Transplanting date	19 October 2018	1 April 2019
Last harvest date	30 January 2019	30 June 2019
Total duration days	104	91
Average Temperature (°C)	21.5 ± 2.6	26.4 ± 2.4
T Max (°C)	27.6 ± 3.0	31.4 ± 3.0
T Min (°C)	17.7 ± 2.8	23.2 ± 2.3
RH Max (%)	82.5 ± 7.0	88.1 ± 5.5
RH Min (%)	54.3 ± 10.0	64.7 ± 12.6
Precipitation (mm)	5 ± 0.2	2404.8 ± 15.3
Season	Winter/dry	Summer/rainy

## Data Availability

The data presented in this study are available on request from the corresponding author. The data are not publicly available for a certain period of time and later can be accessed from https://worldveg.tind.io/.

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
