# Peer review of "Evaluation of Different Bacterial Wilt Resistant Eggplant Rootstocks for Grafting Tomato"

_plants, 2021, doi:10.3390/plants10010075_

Round 1

Reviewer 1 Report

This article is a solid scientific paper with novelty value. Really regrettable that that there was no yield in the second season. Any ideas/conclusions for management (well known as difficult and expensive) for the yellow leaf curl diseases, which is transmitted by whitefly Bemisia tabaci, in this kind of field experiments?

Author Response

Point 1: This article is a solid scientific paper with novelty value. Really regrettable that there was no yield in the second season. Any ideas/conclusions for management (well known as difficult and expensive) for the yellow leaf curl diseases, which are transmitted by whitefly Bemisia tabaci, in this kind of field experiments?

Thank you very much for the review. We used a variety, which is reported to possess resistance to Tomato Leaf Curl Virus disease. However, various species/isolates/strains have been reported as tomato leaf curl disease (ToLCD)-related viruses, and hence the resistance of a tomato variety depends on the Ty resistant gene(s) and the type of virus predominantly occurring in a geographical location. It has been shown that combinations of Ty genes provide better control against begomoviruses than do single genes. Hence, an appropriate resistant tomato variety suitable to overcome the predominant ToLCD causing virus in the experimental location should be selected as the scion for future studies.

Reviewer 2 Report

Manickam et al manuscript titled “Evaluation of different bacterial wilt resistant eggplant rootstocks for grafting tomato” tried 5 bacterial wilt resistant eggplant rootstocks for grafting tomato variety Victoria at season 2018 and variety TStarE at season 2019. Characterized grafting efficiency in both seasons, however many fruit traits were validated in the season 2018 and not in 2019.

Line no 20-22: Here better to include data values also like low wilting percentage (xx%)

Line no 215: please correct to VI046095

Line no 233: sub heading 4.2 complete the heading as Seedling production, grafting and pathogen inoculation or page no 245 to 256 make it separate paragraph heading as Ralstonia pseudosolanacearum strains and inoculation

Line 209 better to use to bulk up instead of increased

Line 259: 18 plants per replication please mention here how many replicates (repeats)?

Line 264: 12 plants per plot please mention here how many replicates (repeats)?

Line 291: per plant three SPAD values were taken and how plants were taken per plot, it must require to 3 or more plants per plot.

Line 201: section 4.1. Plant materials and experimental location please try to mention how much irrigation applied if applied to the irrigation in 2018 and 2019 because 2018 rain fall 5 mm.

Please try to discuss the large scale (field level for farmer) grafting practicality.

Author Response

Thank you very much for the review.

Point 01:

Manickam et al manuscript titled “Evaluation of different bacterial wilt resistant eggplant rootstocks for grafting tomato” tried 5 bacterial wilt resistant eggplant rootstocks for grafting tomato variety Victoria at season 2018 and variety TStarE at season 2019. Characterized grafting efficiency in both seasons, however many fruit traits were validated in the season 2018 and not in 2019.

In 2019, the plants were severely affected by viral leaf curl diseases and the yield was very low. Not enough fruit samples were harvested for analysis. Moreover, the fruit characters were influenced by the disease.

Point 02:

Line no 20-22: Here better to include data values also like low wilting percentage (xx%)

Data included for graft compatibility, wilting percentage and disease index.

All the rootstocks showed good graft compatibility (93% and above) and grafted plants showed low wilting percentage (0.0-20.0%) and disease index (0.0-20.8%) following inoculation with BW

Point 03: 

Line no 215: please correct to VI046095

Corrected in the manuscript.

Point 04:

Line no 233: sub heading 4.2 complete the heading as Seedling production, grafting and pathogen inoculation or page no 245 to 256 make it separate paragraph heading as Ralstonia pseudosolanacearum strains and inoculation

4.2 Sub heading changed to “Seedling production, grafting and pathogen inoculation” as suggested.

Point 05:

Line 209 better to use to bulk up instead of increased

Corrected in the manuscript as “subsequently the accessions were purified and bulked up through single-seed descent”

Point 06:

Line 259: 18 plants per replication please mention here how many replicates (repeats)?

Four replications, included in the manuscript.

Point 07:

Line 264: 12 plants per plot please mention here how many replicates (repeats)?

Four replications, included in the manuscript.

Point 08:

Line 291: per plant three SPAD values were taken and how plants were taken per plot, it must require to 3 or more plants per plot.

All the survived plants in the entire treatment were observed for the SPAD values.

Point 09:

Line 201: section 4.1. Plant materials and experimental location please try to mention how much irrigation applied if applied to the irrigation in 2018 and 2019 because 2018 rain fall 5 mm.

We haven’t measured the amount of irrigation applied for the experiments.

Point 10:

Please try to discuss the large scale (field level for farmer) grafting practicality.

Revised and included this point in the conclusion.

Reviewer 3 Report

Bacterial wilt is hard to control in field and resistance breeding is the most promising method to prevent the disease. However, grafting provides an alternative way to control this disease. The authors have tried different rootstocks and compared their compatibility with scion, disease incidence, yield and quality with resistant check and non grafting controls. The methods and design are adequate, results were well analyzed.

minor concerns:

  1. TstarE was used as Sicon because of resistance to leaf curl virus, but the second season failed because of leaf curl. Is that right?
  2.  Did the author have data of yield for grafted and controls under unchallenged condition? Then it’s clear to see the effects on total yield.

Author Response

Bacterial wilt is hard to control in field and resistance breeding is the most promising method to prevent the disease. However, grafting provides an alternative way to control this disease. The authors have tried different rootstocks and compared their compatibility with scion, disease incidence, yield and quality with resistant check and non grafting controls. The methods and design are adequate, results were well analyzed.

Thank you very much for your review.

minor concerns:

Point 01:

1.TstarE was used as Sicon because of resistance to leaf curl virus, but the second season failed because of leaf curl. Is that right?

For the first season, we used the scion cv.Victoria which is not resistant to the tomato leaf curl disease (ToLCD), but it was not the conducive season for leaf curl disease. For the second season, we anticipated that leaf curl disease was likely to be prevalent and so we used the leaf curl resistant scion cv. TstarE. However, the resistance reaction of a tomato variety depends on the Ty gene(s) in it and the type of ToLCD-virus types occurs in a geographical location; hence, the entire experimental plots were severely affected by leaf curl in the second season.

Point 02: 

 2. Did the author have data of yield for grafted and controls under unchallenged condition? Then it’s clear to see the effects on total yield.

We don’t have the yield data for the unchallenged disease condition, since the entire experiments were conducted in the sick plots in order to have the maximum disease pressure on the resistant root-stocks and the grafting combinations.

Round 2

Reviewer 2 Report

Thank you for the revision and consideration of the suggestions.

Please include in manuscript also what you responded in point 08 and 09.

Point 08:

Line 291: per plant three SPAD values were taken and how plants were taken per plot, it must require to 3 or more plants per plot.

All the survived plants in the entire treatment were observed for the SPAD values.

Point 09:

Line 201: section 4.1. Plant materials and experimental location please try to mention how much irrigation applied if applied to the irrigation in 2018 and 2019 because 2018 rain fall 5 mm.

We haven’t measured the amount of irrigation applied for the experiments.

For this authors can include like intermittent irrigation was applied whenever required or if you know frequency like gap of 2or 3 weeks once for year 2018 and also mention the irrigation information for 2019. 

Author Response

Extremely sorry to miss out on the two points in the manuscript. Thanks for indicating. 

Point 08:

Added in the manuscript as per the recommendation.

Point 09:

Since we don't have the irrigation schedule, I added "Intermittent irrigation was applied whenever required" as per the recommendation.

Revised manuscript attached.